# Eco-Friendly Lead-Free Solder Paste Printing via Laser-Induced Forward Transfer for the Assembly of Ultra-Fine Pitch Electronic Components

**DOI:** 10.3390/ma14123353

**Published:** 2021-06-17

**Authors:** Marina Makrygianni, Filimon Zacharatos, Kostas Andritsos, Ioannis Theodorakos, Dimitris Reppas, Nikolaos Oikonomidis, Christos Spandonidis, Ioanna Zergioti

**Affiliations:** 1Physics Department, Zografou Campus, National Technical University of Athens, 15780 Athens, Greece; marinam@mail.ntua.gr (M.M.); fzach@mail.ntua.gr (F.Z.); kandritsos@mail.ntua.gr (K.A.); jtheod@mail.ntua.gr (I.T.); 2Prisma Electronics SA, 68132 Alexandroupolis, Greece; dimitris.reppas@prismael.com (D.R.); n.oikonomidis@prismael.com (N.O.); c.spandonidis@prismael.com (C.S.)

**Keywords:** Laser-Induced Forward Transfer, lead-free solder paste, PCB assembly, printed micropatterns

## Abstract

Current challenges in printed circuit board (PCB) assembly require high-resolution deposition of ultra-fine pitch components (<0.3 mm and <60 μm respectively), high throughput and compatibility with flexible substrates, which are poorly met by the conventional deposition techniques (e.g., stencil printing). Laser-Induced Forward Transfer (LIFT) constitutes an excellent alternative for assembly of electronic components: it is fully compatible with lead-free soldering materials and offers high-resolution printing of solder paste bumps (<60 μm) and throughput (up to 10,000 pads/s). In this work, the laser-process conditions which allow control over the transfer of solder paste bumps and arrays, with form factors in line with the features of fine pitch PCBs, are investigated. The study of solder paste as a function of donor/receiver gap confirmed that controllable printing of bumps containing many microparticles is feasible for a gap < 100 μm from a donor layer thickness set at 100 and 150 μm. The transfer of solder bumps with resolution < 100 μm and solder micropatterns on different substrates, including PCB and silver pads, have been achieved. Finally, the successful operation of a LED interconnected to a pin connector bonded to a laser-printed solder micro-pattern was demonstrated.

## 1. Introduction

The broad field of microelectronics has rapidly grown over recent decades, owing to novel materials and advanced fabrication technologies, which offer key-solutions to specific challenges. These technologies have also fostered the advancement of die-attach and chip-bonding techniques, which are essential for the packaging and assembly of microelectronic components. Conventional chip-bonding technologies spanning wire or thermosonic bonding and flip-chip bonding, have been improving in terms of throughput and reliability [1], but still cannot process flexible dies [2], nor accommodate the conformal attachment of dies on-chip with non-planar surface morphologies [3,4]. However, novel integration schemes and printable materials [5] have enabled innovative configurations, such as flexible and stretchable devices, aligned with environmental concerns for greener approaches [6]. These integration schemes, incorporating heterostructures and 3D architectures in many cases [7], have laid the foundations for a novel paradigm in interconnection technology: the digital and drop-on-demand fabrication techniques [8]. Demonstrations of the on-demand fabrication of interconnections have matched or even outperformed the conventional bonding technologies over the past 5 years. In particular, the inkjet printing of high-resolution bumps with 2.5 Ω/bump resistance [9] and micro-dispensing with high precision and a vast range of deposited droplet volumes [10] have been demonstrated. Moreover, aerosol jet printing has delivered reliable Au and Al wire bonds [11], and more complex micro-dispensing schemes have been demonstrated for addressing hybrid bonding challenges [12]. Among these drop-on-demand digital technologies, laser direct printing has shown significant promise for breakthroughs in the high-resolution and digital deposition of materials interesting for die attachment and assembly applications: recently, laser direct printing of metal inks and pastes has shown outstanding potential for the fabrication of ultra-fine pitch [13,14], conformal [15] and even flexible interconnections [16]. The laser direct printing of highly viscous Ag nanopaste has been reported in [17] and [18] for the digital fabrication of 3D interconnects and side-wall contacts, respectively. Moreover, laser direct printing is fully compatible with lead-free and printable materials, enabling low temperature processing [19].

Within the family of laser direct printing technologies, laser printing relying on the laser-induced forward transfer (LIFT) technique stands out. LIFT has been progressing since its first demonstration in the early 1970s [20,21] and recently has been established as a direct printing method offering high-resolution and enabling direct deposition of features with minimized involvement of solvents and waste fluids (down to pL volumes). In particular, 20 μm resolution at the liquid phase and around 3 μm and 0.33 μm at the solid phase using nanosecond (ns-LIFT) and femtosecond (fs-LIFT) pulsed lasers, have been demonstrated in [22] and [23] respectively. Over the years, it has been successfully employed as a digital and high-resolution tool for the fabrication of flexible circuits, organic devices and sensors [24]. Contrary to the conventional deposition processes, in which multiple, costly and time-consuming steps are involved, LIFT is a direct printing technique offering digital control over the size and the shape of the transferred voxel [25]. In this respect, LIFT is fully aligned with the trend for greener fabrication technologies, by being both a digital and an additive manufacturing process. The digital character enables significant cost saving without the need for masks and minimal non-recurring engineering times. The additive character fosters energy saving by minimizing the material waste. Moreover, LIFT is not limited by the viscosity of the material under transfer, as in the case of inkjet printing, since it is compatible with viscosities ranging from 1 up to more than 200,000 cP [14,26]. Therefore, it is compatible with both the environmentally friendly metal nanoparticles inks and the more viscous lead-free pastes. Combinational schemes of LIFT and laser micro-sintering in a single process chain have been previously introduced [27], resulting in metallic micropatterns for a large range of applications in flexible electronics including: the printing of flexible radio-frequency identification (RFID) antennas [28], the controllable and reproducible printing of microdots of silver (Ag) nanopaste (viscosity > 100,000 cP) on vertical sidewalls [29] for high performance electrical interconnections and recently the accomplishment of flexible interconnections [30]. Additionally, LIFT is capable of transferring pixels [31,32] and stacks of materials [33,34] in the solid phase [35], as well as intact dies or devices [36], and recently solder paste [37].

Here, we highlight the LIFT’s potential for the digital and high-resolution (sub-100 μm) printing of a type 5, lead-free, solder paste by investigating the optimal process parameters on three different substrate cases, which represent most substrates involved in the die-attach processes: silicon (Si), polymers and metallic PCB pads. Contrary to previous reports on the printing of solder paste bumps [14,37], in this work we demonstrate the actual bonding process by providing a proof of concept. In this respect, the methodology is structured to facilitate the solder paste bonding enabled by LIFT: (i) the donor’s thickness is sufficiently high to ensure that the volume of the transferred solder paste is comparable to the volumes deposited by conventional processes. (ii) The selected receiver substrates are actual gold (Au)/nickel (Ni)/copper (Cu) PCB pads, validating the compatibility with the PCB-assembly technology. (iii) We also demonstrate the versatility offered by LIFT, in terms of form factors, by printing micropatterns consisting of many solder paste bumps, to achieve controllable coverage of the effective area of the PCB pads. (iv) Using LIFT of Ag inks, a test platform comprising pads and electrodes was fabricated; solder paste micropatterns are deposited by LIFT on the designated pads, and then heated to achieve the bonding with the pin connectors which are in turn interconnected with a commercially available LED, whose operation is validated by applying the nominal voltage.

## 2. Materials and Methods

The configuration employed for these experiments consisted of two subsystems, the laser printing and the high-speed imaging system. The former comprised a ns diode-pumped solid-state (DPSS) Nd:YAG laser (InnoLas Photonics GmbH, Krailling, Germany), operating at a wavelength of 532 nm, providing 20 W maximum output power and up to 500 kHz repetition rate, implementing the LIFT method (Figure 1) [38]. A repetition rate range of 10 kHz was employed in this study and the corresponding pulse duration was 20 ns, while the maximum output power used was 1.84 W. In all experiments the laser spot size, which refers to the diameter of the spot size on the donor/carrier interface, was 100 μm. To print more complex patterns than single bumps, the laser beam was scanned with speeds up to 3 m/s by using a galvanometric scanning system and an f-theta lens implementing a focal length of 170 mm. The high-speed imaging system comprised a high-speed camera (Mini AX-100, Photron by Photron Europe Limited, Buckinghamshire, UK) coupled to the printing system, providing a maximum recording speed of 540,000 fps for this model. For these experiments, a recording speed of 122,400 fps (one frame every 8.17 μs) was found to be sufficient for the needs of the investigation on the jet formation and propagation. To illuminate the process, a commercially available light emitting diode (LED) was used, placed opposite to the high-speed camera and focused on the jet’s formation plane, perpendicularly to the donor-receiver substrates. The camera was coupled with a 3× lens, operating as a magnification system. For the accurate calculation of the jet’s velocity, a donor-receiver gap of more than 300 μm was selected.

For the investigation of the ejection mechanism, a sacrificial layer was added to the donor material: a silver nanoparticle (Ag NP) ink layer was used as a dynamic release layer (DRL). Based on recently published findings [37], the use of DRL-assisted LIFT results in more reproducible, homogeneous and geometrically defined deposits, with respect to DRL-free LIFT when solder paste is involved. In particular, the main difference between DRL-free and DRL-assisted LIFT is the transfer mechanism of the solder paste. In the case of DRL-free LIFT, a voxel of solder paste separates from the donor and travels towards the receiver. Conversely, in the case of DRL-assisted LIFT and if the donor/receiver gap is in the order of 100 μm or less, then a jet is formed which travels towards the receiver at a lower laser fluence ejection threshold, resulting in smoother depositions. Therefore, in this study, DRL-coated donors were employed. The DRL was prepared by spin coating a Ag nanoparticle ink (SunChemicals, 20 wt.% silver content) onto the donor substrate (quartz window, 50 mm dia × 3mm thick purchased from UQG Optics) at 1000 rpm for 30 s. After the spin coating procedure, the Ag NP ink was dried at 75 °C for 10 min and then sintered at 175 °C for 10 min, resulting in a thin (300 nm thick) Ag layer [6]. Silver was chosen as the DRL material for compatibility reasons, considering that the solder paste already contains a small percentage of silver. Finally, the solder paste donor layer was formed by coating thin films of the printable solder paste using an adjustable micrometer film applicator, whose blade height was set to four different levels: 15, 50, 100 and 150 μm. Profilometry was used to measure the surface roughness in the four different donors. 

The average roughness (*R_a_*) was calculated using the following formula: (1)Ra=1L∫0L|Z(x)|dx,
where Z(x) corresponds to the profile ordinates of the roughness profile and L to the sampling length. *R_a_* was found to be 5.31 μm, 5.24 μm, 5.04 μm and 3.77 μm, for donor thickness of 150 μm, 100 μm, 50 μm and 15 μm, respectively. The quartz donor substrates used in this study feature very high transmission at 532 nm (>95%) and very low surface roughness (<2 nm). 

For the receiver substrates, two different materials were chosen which are of interest for PCB packaging and assembly technology: semiconductors and polymers. For the printing investigation of discrete bumps, Dupont^®^ Kapton^®^ polyimide (PI) film (25 μm thickness) was chosen, due to its unique combination of mechanical and thermal properties and Si because of its establishment as a processing material in the manufacturing field. Furthermore, the metal pads (Au/Ni/Cu) of a PCB were preferred as the receiver substrate for the printing of a solder paste design. According to the manufacturer (Eurocircuits), typical thicknesses of the metals comprising the PCB pads are within a range between 3 and 6 μm for Ni, 0.05 and 0.12 μm for Au. 

Lead-free, no-clean solder paste, designed for use in Jet Printers, was purchased from Alpha Assembly Solutions. The aforementioned solder paste exhibits properties that are listed in Table 1. Briefly, solder paste consists of metallic microparticles suspended in a flux. Solder paste is categorized by type with regards to its solder particle size, and for example in type 5, 80% of the particles are between 15 to 25 μm. In general, type 5 gives higher transfer efficiency than type 4 solder paste (particle size 20–38 μm) and enables the enhanced printing for miniaturized components. All printed samples underwent a post-printing heating process, by oven-curing between 220 °C for 15 min with a cooling time to room temperature of 30 min, temperature indicated by the solder paste supplier.

The Ag nanoparticle (NP) highly viscous ink employed for the laser printing and laser sintering of electrodes and pads, was procured from P.V. Nano Cell Ltd. (Migdal Ha’Emek, Israel) and consists of solid content, Sicrys™ nano silver particles, in the range 70–75 wt.% . The realized test platform with this ink will be used for the proof of concept.

## 3. Results and Discussion

The compatibility of LIFT with the selected solder paste material and their performance in key-challenges related to the PCB-assembly technology were investigated by conducting three different experimental studies: (i) the influence of the donor layer’s thickness on the jet dynamics, (ii) the influence of the donor/receiver’s gap on the printing quality and (iii) the contributions of the substrate. This section reports on the results derived from these three studies.

### 3.1. Influence of Donor Thickness on Jet Dynamics

An important parameter defining the laser printing quality is the donor thickness. To investigate the influence of the donor thickness on the jet propagation, frames extracted from the videos acquired with the high-speed camera setup at 122,400 fps were processed. The donor-receiver gap was set at 500 μm for the 15 μm and 50 μm thick donors, at 350 μm for the 100 μm thick donor and at 600 μm for the 150 μm thick donor, to extract a sufficient number of frames for the velocity calculation. Figure 2a shows a sequence of frames illustrating the solder paste’s propagation for each donor case at their respective laser fluence transfer threshold. It is evident that the ejected solder paste breaks into smaller fragments until it reaches the receiver in the case of the thinner donors, while a more concrete bump propagates towards the receiver for the thickness of 100 μm. In the case of the 150 μm donor, the donor-receiver gap is quite large (600 μm) and as a result the ejected solder bump is multi-fragmented before reaching the receiver. In Figure 2b the ejection velocities at laser fluences of 0.7 and 0.8 J/cm^2^, 1.0 and 1.2 J/cm^2^, 1.4 and 1.5 J/cm^2^ and 1.7, 2.0 and 2.3 J/cm^2^ are presented for the four different solder paste film thicknesses (15, 50, 100, 150 μm) on the donor. The term “ejection velocity” describes the average jet front velocity. More specifically, the velocity was calculated by a linear fit of the propagation length of the jet front before impact, as a function of the elapsed time for each successive frame extracted from the videos. As expected, the velocities increase as the laser fluence increases for each case. Furthermore, the increase in solder thickness on donor results in a shift of the threshold ejection fluence to higher fluences and towards lower average velocities of the travelling jet.

In general, excessive amount of solder paste on a PCB pad may cause solder bridging, whereas insufficient solder paste may lead to inadequate solder wetting between all components. In the case of stencil printing, a typical thickness of the applied solder is between 100 to 150 μm. Therefore, for consistency reasons, in this study we opted for the thickness of 100 μm layer on the donor to be used in the investigation of the laser printing conditions for the individual printed bumps and a thickness of 150 μm layer on the donor was used for printing solder paste micropatterns on PCB pads and Ag pads.

### 3.2. Influence of the Donor-Receiver Gap

In addition to the jet propagation study as a function of donor thickness, one other important parameter for successful printing is the understanding of the transfer mechanism of solder paste with respect to the donor/receiver gap. In this context, high-speed imaging experiments were carried out at different donor/silicon receiver gap distances. For these experiments two donor thicknesses were used, 100 μm and 150 μm. Figure 3 shows still frame images illustrating solder paste transfer mechanism for each donor/receiver gap distance at their respective laser fluence transfer threshold (1.4 J/cm^2^ and 1.7 J/cm^2^ in the case of 100 μm (Figure 3a) and 150 μm (Figure 3b) thick solder paste donor, respectively). More specifically, in both cases a bridging is observed between the donor and the receiver substrate for donor/receiver gap up to 100 μm. In this bridging-based transfer, the formed solder paste jet continuously feeds the receiver for a period of more than 40 ms or until the donor is removed. The resulting printed bumps are of circular shape with a diameter between 50 to 60 μm (as shown in Figure 3a,b bottom row). In contrast, in the case of donor/receiver gap > 100 μm the solder paste transfer breaks into filaments/smaller fragments before reaching the receiver resulting in a multi-fragmented printed solder bump (as shown in Figure 3a,b bottom row).

It is concluded that the bridging effect that occurs for a donor-receiver gap ≤ 100 μm is beneficial for the process due to the resulting more directional transfers, as opposed to larger gaps where the solder jet may oscillate or break during transfer. Consequently, in this work, the donor-receiver gap was fixed at 50 μm, for optimal printing result. 

### 3.3. Printing on Si and Flexible Substrates

The following results demonstrate the applicability of LIFT for the printing of bumps on widely used semiconductor and polymer substrates, in particular Si and Polyimide (as shown in Figure 4). At the optimized processing conditions (i.e., use of laser fluence just over transfer threshold, laser spot size 100 μm), a microarray of bumps was obtained on Si (Figure 4a) and polyimide (Figure 4b) substrate, for a laser fluence of 1.4 J/cm^2^. It is observed that the flux residues cover the enclosed microparticles of the printed bumps shortly after printing, which are barely visible. The printed bumps on Si keep a round shape (Figure 4a), whereas the control over the printed bumps’ shape on polyimide is proven more challenging with less uniform resulting shapes (Figure 4b). Polyimide has a very low surface energy in the order of 50 mJ/m^2^ [39] and consequently, poor wetting behavior and adhesion is expected with respect to the higher surface energy Si substrate [40].

SEM characterization provided additional information about the microparticle morphology and form factors for both substrate cases, after oven-curing. In Figure 5, the bumps printed using the optimal laser fluence ranges for both Si and polyimide are presented. For increased fluence, the total volume of the bump and therefore the number of enclosed microparticles increases. The microparticle average diameter, is consistent in all cases, ranging from 12–22 μm, in partial agreement with the solder paste’s datasheet. In addition, the flux is almost completely removed owing to the heating occurring during the oven-curing procedure. 

### 3.4. Printing on PCB Pads

Finally, the laser printing of solder bumps on a Au/Ni/Cu pad was investigated towards the direction of demonstrating that the reported process is fully compatible with the widely used pad materials in PCB technology. For the fabrication of patterns on the PCB pads, the donor-receiver gap was fixed at 50 μm and the laser beam was scanned over the donor surface creating multiple jets, which form overlapping bumps (Figure 6a). First, a circular shape with length of 800 μm and filled with five hatched lines was designed and fed to the scanning system software. The scanning speed was set at 1.5 m/s, yielding 16 laser pulses in the perimeter of the circle and 5 laser pulses in the central hatched line. The resulting printed pattern had a circular form of length similar to that of the circular design (Figure 6b) and an average height of 140 μm (Figure 7a). Each bump was printed from a laser spot diameter of 100 μm, at a laser fluence of 2 J/ cm^2^ resulting in average bump diameter of 165 μm. Consequently, the circular solder pattern covers around 50 % of the PCB pad.

Second, to maximize the coverage on the pad and the effective area available for bonding, a square pattern with length of 1100 μm and filled with seven hatched lines was designed and fed to the scanning system, yielding a square solder micro-pattern as shown in Figure 6c. It must be noted that each bump has a diameter of 165 μm and the digital control offered by LIFT ensures that the resulting covered printed area does not overflow the metallic pad’s length of 1100 μm. The solder micro-pattern has an average thickness of 135 μm and length around 1000 μm (Figure 7b), offering around 90% coverage, according to image analysis (area fraction) using ImageJ software.

From Figure 6b,c and Figure 7b, it can be observed that the profile of the upper part of the micropatterns collapses to a lower level of the order of 30–50 μm. This can be attributed to the thixotropic nature of the solder paste material, where upon application of shear stress (caused by the laser pulse) [41] its viscosity decreases. Consequently, during impact of the solder bump on the receiver, the transferred microparticles may rearrange before the solder paste fully recovers its nominal viscosity value.

### 3.5. Proof of Concept: Demonstration of LED Operation

To demonstrate the bonding process enabled by LIFT, a commercially available LED was interconnected with pin connectors bonded to solder paste micropatterns which had been printed on silver pads. More specifically, the process for the realization of the test platform is described in the following steps. First, linear patterns and pads were laser-printed on a glass receiver substrate, as shown in Figure 8a. Prior to the LIFT experiment, the Ag NP ink was coated via an adjustable micrometer film applicator on a transparent carrier to form a 10 μm thick donor substrate. Both the linear electrodes and the pads were fabricated using LIFT, by printing consecutive droplets with a scanning speed of 0.5 m/s and a repetition rate of 10 kHz, at a donor/receiver gap distance of 50 μm. The process was duplicated in the case of the linear patterns to avoid any open circuits. Second, selective laser sintering was carried out by irradiating the printed linear patterns and pads with a scanning speed of 0.1 m/s and a repetition rate of 60 kHz, which resulted in a 98% pulse to pulse overlap, while in the vertical to the printed linear patterns’ direction, a 10 μm distance between successive scanning passes was used, resulting in typical resistivities < 10 × bulk Ag [27]. The outcome of this process is the evaporation of the ink’s solvent and additives and the transformation of the printed nanoparticle pattern into a solid metal track, with significantly increased grain size. For the LIFT process, the laser fluence was set to 300 mJ/cm^2^ and the spot size of the laser beam was 40 μm, and for the sintering process the laser fluence was set at 210 mJ/cm^2^, and the spot size of the laser beam was 100 μm. The resulting Ag patterns are all conductive and their form factors were measured using profilometry: the pads have an average thickness of 1.6 μm and length around 2000 μm (as shown in inset of Figure 8a); the linear electrodes have an average thickness of 3 μm and width around 190 μm (as shown in inset of Figure 8b).

Following the formation of electrodes, LIFT of solder paste square micropatterns was carried out on the electrode pads. For this, a square shape pattern with length of 1500 μm and filled with 10 hatched lines was designed and fed to the scanning system software. The scanning speed was set at 1.5 m/s, the donor/ receiver gap was fixed at 50 μm, laser spot diameter at 100 μm, and laser fluence at 2 J/cm^2^. The process was replicated to stack one more micro-pattern on top of the first. The resulting micro-pattern has an average thickness of 220 μm and length around 1300 μm (as shown in inset of Figure 8c).

Next, 3 pin connectors were embedded on the solder micropatterns prior to the thermal processing step described in Section 2 (Figure 8c). Successful bonding between the connectors and the solder micropatterns was observed after the oven-curing processing step. Finally, a LED was interconnected to the bonded pin connectors. To operate the LED, 3 V was applied on the silver pads via a Keithley 2400 current source. The successful operation of the LED was proven as shown in Figure 8d.

## 4. Conclusions

This work demonstrates that LIFT is compatible with lead-free solder paste and die attachment technology within a wide process window and for at least three different receiver materials: silicon, PI and Au/Ni/Cu metal. An investigation on the solder paste printing as a function of laser fluence confirmed that controllable printing of bumps containing many microparticles is feasible from a donor layer thickness set at 100 μm on Si receiver substrate. Printing was also achieved for polyimide receiver substrate, despite its low surface energy value. For all cases, oven heating according to the solder paste specifications led to flux removal, exposing the metal microparticles. The laser printing process development was applied for the successful fabrication of millimeter sized solder paste patterns on top of PCB pads with sufficient surface coverage to ensure bonding. Finally, the successful operation of a LED was validated. The LED was interconnected to pin connectors bonded to laser-printed solder paste micropatterns on a test platform consisting of laser-printed and laser sintered Ag NP highly viscous ink.

## Figures and Tables

**Figure 1 materials-14-03353-f001:**
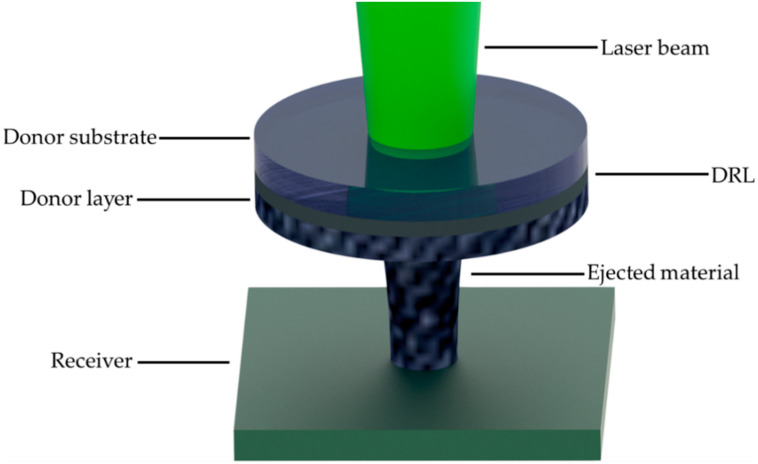
Schematic representation of the LIFT process of solder paste.

**Figure 2 materials-14-03353-f002:**
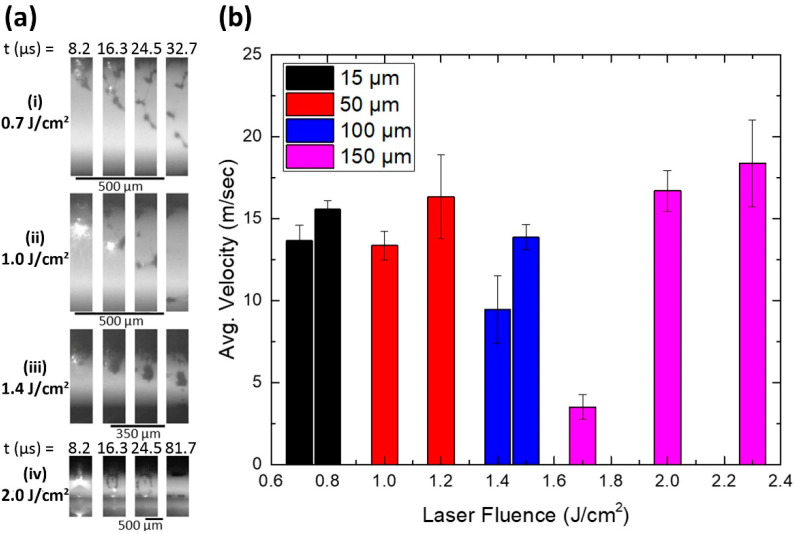
(**a**) Sequence of frames extracted from videos acquired at 122,400 fps for solder paste propagation from a donor layer thickness set by the film applicator at (**i**) 15 μm and using laser fluence of 0.7 J/cm^2^; (**ii**) 50 μm and using laser fluence of 1.0 J/cm^2^; (**iii**) 100 μm and using laser fluence of 1.4 J/cm^2^; (**iv**) 150 μm and using laser fluence of 2.0 J/cm^2^. The donor-receiver gap was set at (**i**,**ii**) 500 μm, at (**iii**) 350 μm and at (**iv**) 600 μm. (**b**) Ejection velocities as a function of laser fluence for the four different solder paste donor thicknesses: 15 μm (black), 50 μm (red), 100 μm (blue) and 150 μm (magenta). The standard deviation resulting from a set of 10 repeats is also marked.

**Figure 3 materials-14-03353-f003:**
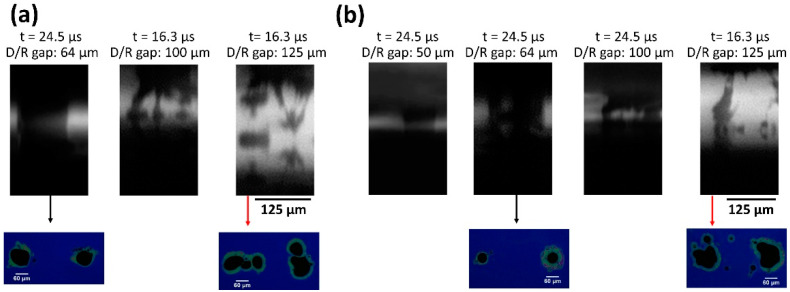
Still frame images extracted from videos acquired at 122,400 fps for a donor/receiver gap set at (**a**) 64 μm, 100 μm and 125 μm (from left to right respectively) and from a donor layer thickness set by the film applicator at 100 μm and using laser fluence of 1.4 J/cm^2^; (**b**) 50 μm, 64 μm, 100 μm and 125 μm (from left to right respectively) and from a donor layer thickness set by the film applicator at 150 μm and using laser fluence of 1.7 J/cm^2^; Bottom row: Optical microscope images of laser-printed bumps on Si substrate at 1.4 J/cm^2^ (**a**) and 1.7 J/cm^2^ (**b**) for donor/receiver gap set at: 65 μm and 125 μm.

**Figure 4 materials-14-03353-f004:**
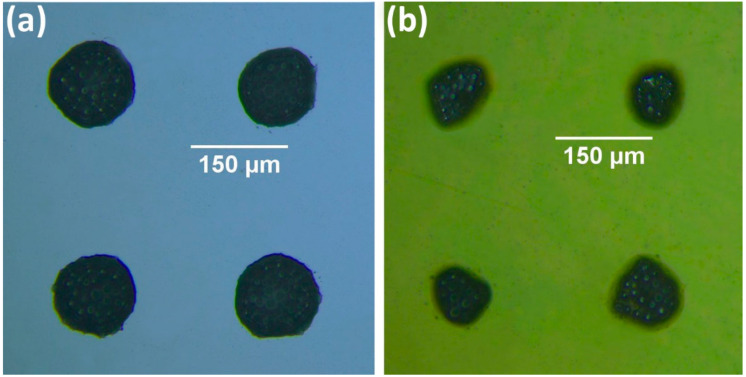
Optical microscope images of laser-printed bumps on (**a**) Si, and (**b**) on polyimide. Laser fluence used for both cases is 1.4 J/cm^2^. Before oven-curing, the flux covers the microparticles which are hardly visible. Donor-receiver gap was fixed at 50 μm and donor layer thickness was set by the film applicator at 100 μm.

**Figure 5 materials-14-03353-f005:**
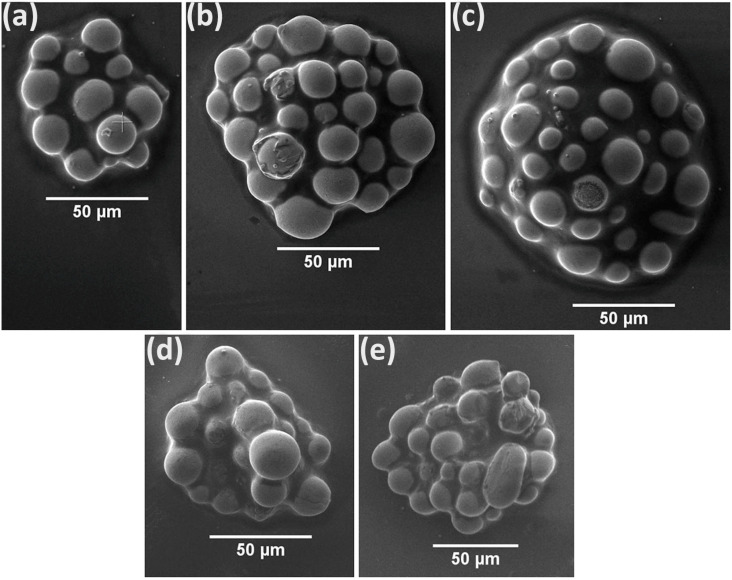
SEM characterization of laser-printed solder bumps post-oven-curing. (**a**–**c**) on Si, 1.3 J/cm^2^, 1.4 J/cm^2^, 1.6 J/cm^2^, respectively. (**d**,**e**) on PI substrate, 1.3 J/cm^2^, 1.4 J/cm^2^, respectively.

**Figure 6 materials-14-03353-f006:**
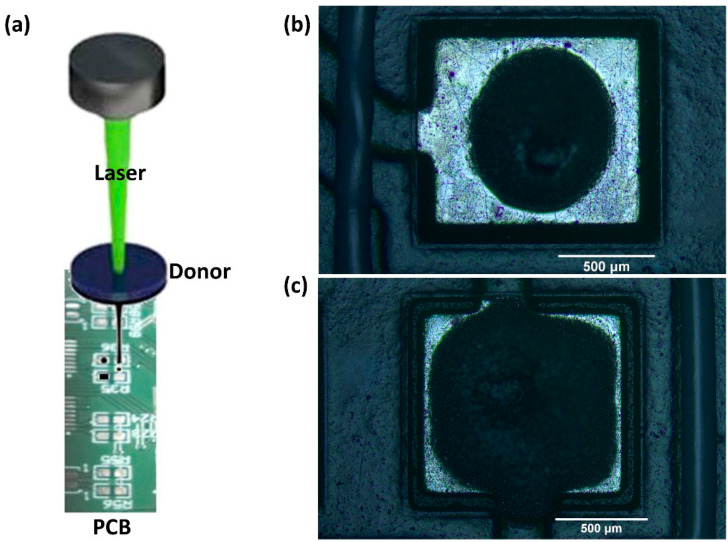
(**a**) Schematic of LIFT printing of solder paste on PCB pads, (**b**) resulting printed solder paste circular pattern at 2 m/s (**c**) resulting printed solder square micro-pattern at 2 m/s. Donor-receiver gap was fixed at 50 μm and donor layer thickness was set by the film applicator at 150 μm.

**Figure 7 materials-14-03353-f007:**
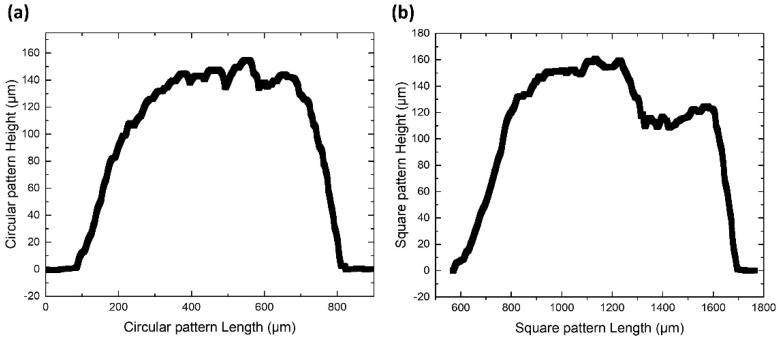
Profilometry measurement of (**a**) a circular printed pattern at 1.5 m/s, and (**b**) a square printed pattern at 1.5 m/s. Donor-receiver gap was fixed at 50 μm and donor layer thickness was set by the film applicator at 150 μm.

**Figure 8 materials-14-03353-f008:**
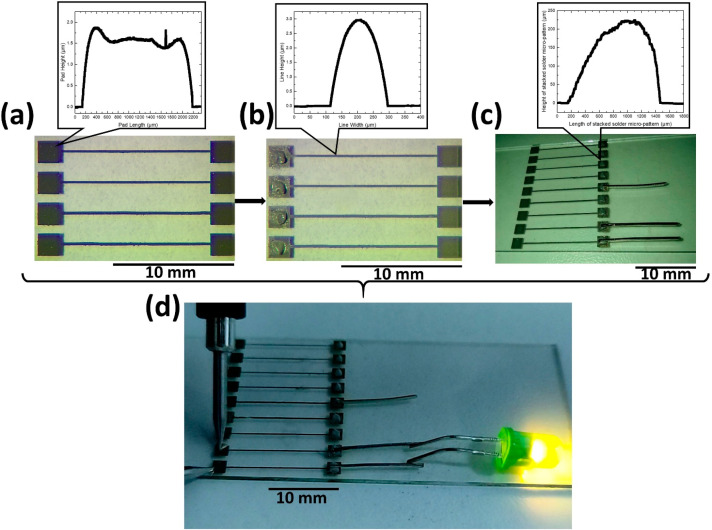
(**a**) Image of laser-printed and sintered linear patterns each connected to their respective pads; (**b**) Image of laser-printed solder paste stacked micropatterns on pads; (**c**) Image of the deposited pin connectors on the solder micropatterns prior to thermal treatment; Insets: Profilometry measurements of the LIFT printed and sintered pad (top left) and linear pattern (top middle) and of the LIFT printed solder paste micro-pattern (top right); (**d**) Operating a commercial LED by applying 3 V on LIFT printed and laser sintered silver pads.

**Table 1 materials-14-03353-t001:** Solder paste properties.

Solder Paste		Type	Particle Size	Category	Flux
ALPHA JP-500	Lead-free, no-clean solder paste for Jet Printing	Type 5	15–25 μm	SAC305 96.5% tin 3% silver 0.5% copper	Zero halogen flux formulation, colorless flux residue

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
