# Peer review of "Eco-Friendly Lead-Free Solder Paste Printing via Laser-Induced Forward Transfer for the Assembly of Ultra-Fine Pitch Electronic Components"

_materials, 2021, doi:10.3390/ma14123353_

Round 1

Reviewer 1 Report

This paper is almost the same as Reference 38. I'm not sure what's different.

Need to explain what's improved.

It is recommended to emphasize the improvement of donor[Ref. 38] by adding DRL-related tests.

The supplementary is interesting.

It would be better to shorten the main text and add supplementary data to the text.

In particular, I hope you discuss the laser sintering process after printing.

[Line 85-88]

Type 5 is a regular standard expression but needs additional explanation, particle size needs to be explained.

[Line 279-300]

Isn't the soldering paste a very viscous material?

A bump is a set of particles caused by stack(collision), so the surface is uneven.

And it doesn’t restore because of its strong viscosity. Besides, there is almost no surface tension.

Is it related to having a flow property?

[supplementary Fig. S2]

LED junction is invisible.

It is suspected that the power line is directly touching the LED terminal.

In actual soldering, terminals touch each other, but it is meaningless to touch terminals directly in this paper that need to test the performance of soldering.

Author Response

Dear Reviewer please find attached our point-by-point response to your comments.

Reviewer 2 Report

The paper feels very similar to their work JLMN-Journal of Laser Micro/Nanoengineering Vol. 15, No. 3, 2020  .

Could the author please highlight the differences .

Author Response

Dear Reviewer please find attached our response to your comment.

Round 2

Reviewer 2 Report

Comment 1: As the author points out this paper  r, [37] is the proof of concept, and presents the results of a parametric experimental study for specifying the process window for solder paste printing using LIFT. Conversely, in the current manuscript the authors investigate the jet dynamics and the laser process conditions which allow control over the transfer of solder paste bumps and arrays, with form factors in line with the features of fine pitch PCBs" I would suggest teh authors to use the exact same words in their abstract instead of claiming "In this work, a novel process for printing lead-free solder paste (powder size 15-25 17 μm) relying on LIFT is reported. "

2. this statement is not clear : " "In this respect, the methodology is structured so as to 108 enable this proof of concept: i) the donor’s thickness is sufficiently high so as to ensure that the volume of the transferred solder paste is comparable to the volumes deposited by conventional processes. ii) The selected receiver substrates, are actual, gold (Au)/nickel (Ni)/copper (Cu) PCB pads, validating the compatibility with the PCB assembly technol-112 ogy. iii) We also demonstrate the versatility offered by LIFT, in terms of form factors, by 113 printing micro-patterns consisting of a large number of solder paste bumps, in order to 114 achieve controllable coverage of the effective area of the PCB pads. iv) Using LIFT of Ag 115 inks, a test platform comprising pads and electrodes was fabricated; solder paste micro-116"
What concept is being proven

3. " To demonstrate the ability of bonding pin connectors using LIFT, a commercially available LED was interconnected to solder paste micro-patterns which had been printed on silver pads. More specifically, the entire test platform realization process is described  in the following steps" The line "More specifically, the entire test platform realization process is described  in the following steps" does not read well

The paper suffers from repetition error for instance "  Second, selective laser sintering was carried out by irradiating the printed linear patterns and pads, with the optimized sintering process parameters for highly viscous Ag NP inks [27], resulting in typical resistivities < 10 x bulk Ag. " In the second line, the optimized parameter of SLS is presented. the two sentences can be combined into one. I would suggest the authors read through the paper carefully and correct the grammar before submitting

Author Response

The authors would like to thank the reviewer for the careful reading and his suggestions that will improve our final manuscript. A point-by-point response can be found in the attached file.
